# Challenges of Gene Editing Therapies for Genodermatoses

**DOI:** 10.3390/ijms24032298

**Published:** 2023-01-24

**Authors:** Imogen R. Brooks, Adam Sheriff, Declan Moran, Jingbo Wang, Joanna Jacków

**Affiliations:** St John’s Institute of Dermatology, King’s College London, London SE1 9RT, UK

**Keywords:** genodermatoses, gene editing, CRISPR, base editing, prime editing, gene editing strategies, off-targets, genetic skin disease

## Abstract

Genodermatoses encompass a wide range of inherited skin diseases, many of which are monogenic. Genodermatoses range in severity and result in early-onset cancers or life-threatening damage to the skin, and there are few curative options. As such, there is a clinical need for single-intervention treatments with curative potential. Here, we discuss the nascent field of gene editing for the treatment of genodermatoses, exploring CRISPR–Cas9 and homology-directed repair, base editing, and prime editing tools for correcting pathogenic mutations. We specifically focus on the optimisation of editing efficiency, the minimisation off-targets edits, and the tools for delivery for potential future therapies. Honing each of these factors is essential for translating gene editing therapies into the clinical setting. Therefore, the aim of this review article is to raise important considerations for investigators aiming to develop gene editing approaches for genodermatoses.

## 1. Introduction

Monogenic genodermatoses encompass numerous genetic skin conditions, including malignancy-causing disorders, blistering disorders such as epidermolysis bullosa (EB), and keratinisation disorders such as ichthyoses [1,2]. These can range in severity from skin rashes that cause pain to life-threatening chronic wounds, cancers, and systemic effects. All are caused by genetic mutations in one or both copies of an allele resulting in abnormal protein function in the skin. In severe cases, there is growing interest in treating the cause of the disease rather than the symptoms, such as via gene therapy.

Gene editing is typically performed using nucleases that recognise specific DNA sequences through either protein structure or through RNA interactions. The nucleases cleave the DNA at this precise locus to cause a double-stranded break (DSB), triggering one of the cellular repair mechanisms for changing the DNA sequences. Broadly, these endonucleases can be categorised by whether protein structures or RNA molecules are used to recognise the specific target DNA sequence. Meganucleases and transcription activator-like (TAL) effector nucleases (TALENs) use protein structures that can be modified to target specific sites [3]. CRISPR–Cas9, on the other hand, uses a single guide RNA (sgRNA) to direct the nuclease to the site of interest containing desirable protospacer adjacent motifs (PAMs). In both cases, DSBs are induced, and these may then be repaired by the cell using non-homologous end joining (NHEJ), which is error-prone but efficient as DNA strands are blunted then ligated together. The other option is homology-directed repair (HDR), which uses template DNA to encode the repair with a low error frequency. Typically, the other chromosome copy is used, but one can introduce an oligonucleotide donor template, through which the edit is installed (Figure 1). Through this mechanism, TALENs and meganucleases have been used to target disease-causing mutations of epidermolytic ichthyosis in keratinocytes, xeroderma pigmentosum, recessive dystrophic EB (RDEB), and EB, to name some examples [4,5,6,7]. CRISPR–Cas9 has been employed to correct pathogenic mutations in dominant dystrophic EB (DDEB), junctional EB (JEB), EB simplex (EBS), RDEB, and epidermolytic palmoplantar keratoderma, among other genodermatoses in fibroblasts, keratinocytes and induced pluripotent stem cells (iPSCs) [8,9,10]. This has resulted in high-efficiency edits, with much lower costs than protein-structure-directed nucleases.

While CRISPR–Cas9 has revolutionised the field of gene editing, inducing DSBs come with risks, especially in therapeutic applications. As such, DSB-free DNA editing strategies are highly desirable. Base editors (BEs) are fusion proteins containing a modified nickase Cas9 (nCas9) that causes single-stranded breaks (SSBs) and a cytidine or adenine deaminase domain [11] called a cytosine base editor (CBE) or an adenine base editor (ABE), respectively. This allows the sgRNA to be used to recognise a specific sequence, as in traditional CRISPR strategies; however, instead of inducing a DSB, a transition mutation is induced within an editing window determined by localisation to the PAM site. BEs are attractive for use in slowly dividing post-mitotic cells such as fibroblasts and keratinocytes because this bypasses the need for HDR, which is less common in post-mitotic cells (Figure 1). Adenine base editors have been trialled in genodermatoses [12,13,14], and cytosine base editors could be used to address many disease-causing mutations.

Prime editors (PEs) are similar to base editors, but a reverse transcriptase is fused to nickase Cas9 (nCas9) instead of deaminase, resulting in a protein that can nick DNA, bind the single-stranded DNA post-nicking, and use an RNA template located in the prime editing guide RNA (pegRNA) to encode an edit onto the DNA [15] (Figure 1). PEs can address all types of mutations and can edit mutations a greater distance from the PAM sequence than base editors, thus demonstrating the wider applicability of prime editing for genodermatoses. Prime editing is a recent development, so it has only been tested for RDEB, resulting in corrections of up to 10.5% [16].

PAM sites limit which mutations can be targeted. As such, there is interest in modifying existing Cas9, base editors, and prime editors to relax the PAM requirements from classical 5′-‘NGG’. Near-PAMless base editors and Cas9 proteins have been developed; these can target most of the genome and have been used to edit human cells [17]. Although they have not been tested in genodermatoses, in silico analysis predicts that an increased number of RDEB-causing mutations can be targeted with PAM-less editors [16].

There are three fundamental approaches to gene editing therapies that may be suitable for genodermatoses. The first is the most commonly explored approach, wherein the variant that causes the genodermatoses is corrected to the wild-type sequence (Figure 2A). This has been demonstrated in in vitro models for EBS, which is caused by mutations in *KRT14* and *KRT5* [18], as well as in JEB [10,19]. This restores functional protein expression, thus treating the disorder. Alternatively, gene editors can be used to install silencing mutations that cause gene knockout (Figure 2B). This can be achieved by installing specific edits to create a premature ‘stop codon’ or through the NHEJ pathway causing indels that result in frameshifts. [20,21]. This approach is advantageous in dominant genodermatoses such as some forms of EBS, DDEB and epidermolytic ichthyosis. Hypermorphic mutations encoding the *KRT10* gene in keratinocytes (causing epidermolysis ichthyoses), have been silenced by NHEJ-induced frameshift mutations [6]. The third approach that is being explored is to target non-variant loci to induce an alternate change to the gene sequence, such as deleting an exon (Figure 2C). This has been explored in RDEB in deleting *COL7A1* exon 73 and 80. These deletions are in frame, so they do not significantly alter protein structure, and they result in potential therapeutic approaches that would benefit many people with variants on the same exon [7,22]. All three of these approaches can be used for the development of gene editing therapies for genodermatoses.

Most gene therapy research to date has focused on gene replacement therapy, using vectors to insert functional copies of the mutated gene into patient cells either in vivo directly onto the skin or ex vivo, where gene-modified cells are therapeutically injected or engrafted into a patient. This has resulted in successful Phase I and Phase II trials in progress for JEB, RDEB, and Netherton syndrome [23,24,25] (Table 1). However, one of the major challenges of gene replacement therapy is insertional mutagenesis, where the gene product is inserted and disrupts gene expression, resulting in the inhibition of tumour suppressors or the activation of oncogenes. This may occur through enhancer or promotor insertion, insertion directly into the gene, or through mutations by mRNA 3′ end substitution [26]. This has resulted in cases such as oncogenesis in patients treated with gene therapy for X-linked severe combined immunodeficiency (X-SCID) [27], and lymphoma has been reported in two patients after chimeric antigen receptor T-cell (CAR-T) therapy [28,29]. Additionally, the non-physiological expression levels of the transgene is a concern in many gene replacement therapies, leading to the toxic overexpression of the therapeutic transgene [30]. This, in turn, has been observed to cause the cell to silence the inserted gene, as seen in some animal studies with transgene therapies [31,32]. As such, there is a growing interest in gene editing tools for addressing genodermatoses.

Gene editing tools directly correct the mutation in the relevant gene, restoring the physiological expression of the gene without the risk of insertional mutagenesis. However, there are still many challenges to consider before gene editing therapies can be applied to genodermatoses in the clinical setting. These include gene editing efficiency, reducing off-target effects, effective delivery, and animal modelling.

## 2. Gene Editing Efficiency

Efficiency in any gene editing approach is crucial in order to restore sufficient protein function to reverse the phenotype. The efficiency of installing precise gene edits with CRISPR-based technologies can be highly variable and is dependent on the technologies used, the sequence of the target DNA, heterochromatin [37], and sgRNA design. Variations exist both between approaches (Cas9/HDR, base editing, and prime editing) and within approaches due to designed differences in protein structures and additional molecules added, which affect the efficiency and specificity of the editing.

### 2.1. Editing Efficiencies Already Achieved in Genodermatoses

Editing efficiencies with CRISPR, base editing, and prime editing can be highly variable and dependent on the specific tool used, cell type targeted, and the mutation. Of the three, CRISPR–Cas9 is the most extensively tested for RDEB, DDEB, and JEB, resulting in up to 94% editing efficiency when cells are selected for GFP/dsRED co-transfection, puromycin resistance selection, or single clone expansion [8,19,38,39,40,41]. This makes CRISPR highly desirable for ex vivo therapies. However, efficiency has typically been lower when selection methods are not used, posing a challenge for in vivo delivery approaches for gene editing therapies.

ABE have also been tested in the context of RDEB. ABEmax was used to achieve editing efficiencies of 23.6 and 30.6% in RDEB patient-derived fibroblasts, as well as successfully editing RDEB-derived iPSCs and restoring functional collagen VII (C7) in human skin equivalents. ABE8e, a novel base editor variant with a 590-fold increased enzymatic activity [42], resulted in 95% editing efficiency in RDEB patient fibroblasts in a bulk population without the need for any GFP selection. This suggests that base editors are potentially powerful tools for therapeutics [43].

Prime editing, as a highly novel tool, has only been used with RDEB, resulting in an editing efficiency of up to 10.5% [16]. This was achieved using PE2, with a nicking guide RNA included to utilise the PE3 system. While prime editing can target many mutations that base editing cannot, there is much optimisation needed before the efficiency of prime editors can be compared to that of base editors.

### 2.2. Improving Gene Editing Efficiency in the Future

Currently, there are new tools available that may increase editing efficiencies in genodermatoses, so they are worth considering for clinical applications. Crucially, efficient guide design is required. For CRISPR–Cas9, many tools have been developed for guide design, performing specificity and efficiency analysis to help choose the best sgRNA. CHOPCHOP is one of the most cited tools for gene editing gRNA design. CRISPOR is also highly recommended, as it also include cloning, validation and expression help [44,45]. For base editing, there are multiple tools such as BE-designer and BEditor, although optimizing for efficiency is more challenging due to the editing window [46,47]. As prime editing is a newer technology, fewer tools are available—Prime Design and PnB Designer have been used to design pegRNAs for prime editor [48,49]. Efficiency prediction in prime editing is more challenging due to the number of variables within pegRNAs [15]. Modified CRISPR–Cas9 proteins that increase the site recognition and cleaving efficiency of the protein, as well as enabling the targeting of novel sites, are regularly researched. One example is CRISPR-PLUS, a Cas9 fusion with recombination J protein, which can increase mutagenesis efficiency by up to 600% [50]. However, one of the main challenges for CRISPR–Cas9-based editing is the need to increase HDR efficiency. One option is to inhibit the NHEJ pathway with small molecules such as SCR7, which prevents ligase IV from binding to DNA and therefore favouring the HDR pathway. This was found to produce a 2-fold increase in HDR-mediated knock-in corrections in porcine foetal fibroblasts [51,52]. However, there is some risk in inhibiting the NHEJ pathway in a therapeutic context. NHEJ inhibition has increased levels of DNA damage, growth inhibition, and cell death in pancreatic tumour and Dalton’s lymphoma cells [53,54]. It is likely that the same impact would be had on non-cancerous cells. Using these tools in a therapeutic in vivo or ex vivo context would need extensive safety testing. An alternative approach, namely enhancing HDR with molecules such as RS-1 that stimulates RAD51, has shown promise for increasing knock-in efficiency in vivo in rabbits by more than two-fold. Similar approaches enhance RAD51 or CtIP at the target locus [55,56]. By avoiding the risk of inhibiting a DNA repair mechanism, it can be speculated that enhancing HDR carries less risk than inhibiting NHEJ; however, this requires further work.

Much like CRISPR, many novel base editing proteins have been developed since their initial publication. The addition of uracil glycosylase inhibitors and nickase activity resulted in the improved cytosine base editors BE2, BE3 [11], and BE4. Codon optimisation further improved the editing efficiency of BE4max by 1.8 fold over its predecessor [57]. When the ABE was first introduced, seven iterations were published in a single paper, resulting in ABE7.10. Later publications optimised codon usage, resulting in ABEmax [57]. The further phage-assisted evolution of the TadA domain resulted in the development of ABE8e that demonstrated 590-fold improved deamination kinetics, resulting in a higher editing efficiency (albeit with higher off-target effects) [42]. The difference in editing efficiency seen in RDEB editing papers using ABEmax compared with ABE8e demonstrates the value of considering novel variants of proteins for improving gene editing efficiency in genodermatoses. Recently, it was also demonstrated that the inhibition of p53 improves base and prime editing efficiency in iPSCs; however, this approach poses challenges in a clinical setting due to the importance of p53 in cell cycle maintenance [58].

In 2021, a new prime editing protein was published—PEmax; this protein has improved codon optimisation as well as additional nuclear localisation signals and mutations in the spCas9 site. This again improved editing efficiency up to 2-fold over PE2. However, PEmax results in higher indels as a consequence of its higher editing efficiency [59]. As mentioned previously, the addition of a second sgRNA to nick the unedited strand can be used to improve prime editing efficiency by up to 5 fold, but with the significant risk of double-stranded breaks, in a system referred to as PE3 [15]. Alternatively, the PE4 system uses a dominant negative form of MLH1 to inhibit the mismatch repair system, with editing efficiency increased by up to 7.7 fold. By combining both PE3 and PE4 approaches in PE5, 1.2-to-2.5-fold editing efficiency improvements were achieved over PE3 [59]. Due to the length of pegRNAs, they are rapidly intracellularly degraded. In a recent publication, this was addressed by engineering pegRNA to contain either a tevopreq1 or an mpknot motif, which stabilised the engineered pegRNA (epegRNA) and prevented degradation [60]. Combining epegRNA and PE5max was found to result in a 12-fold increase in editing efficiency over PE3 [59]. Many of these approaches have not yet been tested in genodermatoses but show promise in addressing diseases caused by point mutations, insertions and deletions.

### 2.3. Improving Efficiency by Selecting for Edited Cells

Editing efficiency can be improved by selecting for edited cells by using either GFP markers or other methods such as antibiotic resistance, which has improved the editing efficiency of CRISPR–Cas9 in DDEB and RDEB by up to 94% [9,38,61]. This approach is also not possible in vivo. Increasingly, other good manufacturing practice (GMP) approaches [62] are being pursued for isolating edited cells, but the secreted nature of C7 makes it hard to select for cells expressing it. One approach may be to edit patient-derived iPSCs. As a colony can grow from a single cell, a colony can be separately grown and sequenced, and a colony that has come from a single edited cell can be selected, resulting in a 100% editing efficiency. These iPSCs can then be differentiated into ex vivo gene therapies. However, this can result in a high genetic drift, increased risk of chromosomal abnormalities, and risks of tumorigenicity [63,64,65,66].

### 2.4. How Much Efficiency Is Really Needed?

While optimising for efficiency is a crucial part of therapeutics, the goal may not be 100% in gene editing. In dominant forms of a disease, the correction efficiency only needs to be high enough that the normal proteins form sufficient complexes with other normal proteins without being inhibited by the dominant negative form. In recessive disorders, one functional copy of a gene is sufficient to restore healthy protein expression. In RDEB, blistering in patients is caused by the loss of function of C7, which is an integral protein in anchoring fibrils (AFs), which ordinarily tethers the epidermis to the dermis. Therefore, the restoration of C7 is critical to reverse the phenotype; however, it is noteworthy that even 15.7% gene correction in primary fibroblasts was shown to be sufficient to recapitulate anchoring fibrils following the grafting of human skin equivalents (HSEs) onto mice [19,67]. Additionally, gene correction can result in disproportionate improvements in mRNA expression [16]; however, this has not been demonstrated in the clinical setting yet.

Compound heterozygosity seen in certain genodermatoses such as RDEB may mean that a higher editing efficiency is needed, as only one allele is being targeted for correction. Multiplexing may be possible to target both alleles for correction in future.

It is possible to optimise gene editing efficiency; however, this goal often has to be balanced with corresponding increases in safety concerns such as off-target editing. Pinpointing the balance between a correction efficiency that can sufficiently restore protein levels but still minimise off-target effects is key to future therapy in genodermatoses.

## 3. Off-Targets

Despite immense potential, the clinical translation of CRISPR–Cas9-based therapy for genodermatoses has been significantly hindered by their propensity to target regions of the genome outside of the intended locus, known as off-target editing. If double-stranded DNA cleavage occurs at the off-target site, stochastic indel mutations via NHEJ or even chromosomal rearrangements can occur. Off-target editing can therefore cause oncogenesis or undesirable changes to gene function that demand an evaluation of off-target events to understand and diminish them. Based on experimental data and sophisticated bioinformatic algorithms, multiple off-target prediction tools have been designed, allowing for not only the selection of gRNAs with favourable safety profiles and properties but also the streamlined and more affordable testing of potential unintended mutations. These are described next.

### 3.1. Interrogating Off-Target Genomic DNA Editing

In recent years, a number of different methodologies have been proposed to analyse off-target changes to genomic DNA post-editing. These can be broadly classed into ‘in-silico’ prediction methods and ‘experimental’ methods [68].

#### 3.1.1. In Silico Prediction Methods

The likelihood of an off-target event occurring at a particular site depends on its similarity to the target sequence and the affinity of the nuclease to this region. Off-target editing is therefore more likely to cluster at loci that share these characteristics, and researchers have described in silico tools that predict these and obviate the cumbersome task of analysing the entire genome. ‘CRISPOR’ [44] employs a validated algorithm known as ‘cutting-frequency determination’ (CFD) [69] to accurately rank thousands of putative off-target sites by those most likely to be edited and allow these sites to be chosen and screened. This approach has been used to analyse base editing off-target effects in RDEB, finding very low off-target analyses [14,70]. The two machine-learning algorithms found to most accurately predict and rank off-target sites are the ‘Elevation’ and ‘CRISTA’ tools [68,71,72]. Another tool, CCtop, has been used to predict off-target sites for the CRISPR–Cas9 editing of RDEB iPSCs [39]. Once predicted off-target regions have been ranked, investigators can then amplify these genomic regions of interest and probe them for off- target editing.

The Surveyor [73] and T7E1 [74] assays—previously used for probing predicted off-target regions in multiple genodermatoses editing studies [61,75]—have now fallen out of favour due to their limitations in detecting small indels [76]. A common approach is to perform the Sanger sequencing of DNA amplicons followed by the analysis of Sanger traces for the insertion or deletion events using the ‘Tracking of insertions, deletions and recombination events (TIDER)’ software [77]. TIDER aligns Sanger chromatographs from edited and unedited samples and determines if indels have occurred as an imprint of aberrant Cas9 cleavage and subsequent NHEJ. TIDER, and the similar bioinformatic tool Inference of CRISPR Edits (ICE) [78], can also detect shifts of nucleotide changes in non-indel-generating off-target editing, such as following base editing.

Although fast and cost-effective, Sanger sequencing methods nonetheless suffer from a poor sensitivity that can leave undetected mutations with a low prevalence in a pool of edited cells [79]. Next-generation sequencing (NGS) techniques, on the other hand, offer gold-standard sensitivity and can be combined with in silico predicted tools to enable the powerful ‘targeted’ NGS sequencing of genomic DNA amplicons. NGS can screen amplicons of predicted off-target sites and detect mutations with a prevalence of less than 0.1% [80], as tested by Sheriff et al. [70]. NGS analysis can reveal background mutations unrelated to off-target activity, which necessitates the sequencing of unedited samples as controls [68].

#### 3.1.2. Experimental Methods

However, all in silico methods share the disadvantage of missing rare unpredicted off-target events [76]. Studies have revealed that SpCas9, which traditionally binds to a ‘5′-NGG-3′ PAM site, has a mild affinity to 5′-NAG-3′ or 5′-NGA-3′ sequences and can also tolerate up to six ‘mismatches’ (six different nucleotides in the protospacer to the target) and still bind [81]. This challenges even the most robust prediction algorithms and explains the finding that selecting the top 10 or 20 predicted off-target sites risks overlooking true off-target editing. To solve this, researchers have described experimental techniques not biased by a priori prediction that can assess the genome in an unrestricted manner for off-target cleavage activity. Digenome-seq [82], one of the first developed techniques, performs the sequencing of purified genomic DNA digested by Cas9 nucleases to find off-target editing with a sensitivity of 0.1%. An updated version, DIG-seq [83], is able to perform this on chromatinic DNA, which better reflects real off-target events in a cell. In comparison, CIRCLE-seq is a method that possesses both a higher sensitivity (0.01%) and a higher signal-to-noise ratio than Digenome-seq and DIG-seq, as off-target fragments are enriched with ligating adapters [84]. CIRCLE-seq, however, requires a greater starting quantity of DNA and suffers from high false-positive rates because it cannot use chromatin as a substrate, unlike DIG-seq. GUIDE-seq is yet another method that detects DSB repair at off-target sites instead of Cas9 cleavage activity like the aforementioned tools [85]. GUIDE-seq incubates cells with double-stranded DNA oligodeoxynucleotides (dsODNs), which are incorporated at Cas9 cleavage sites during repair processes, and has been shown to offer a high sensitivity whilst maintaining low background signals and false-positive rates. These qualities have made it the dominant choice for the experimental evaluation of off-target editing. However, a disadvantage is that not all primary cells respond well to dsODN transfection, which may limit its use in keratinocytes, fibroblasts, and other patient-derived skin cells, although this has not been tested to our knowledge. In these cases, CIRCLE-seq may remain a suitable alternative for sensitive genome-wide off-target evaluation [84], as demonstrated by Osborn et al.; however, in their application, CIRCLE-seq did not identify any of the off-target sites that were edited, as identified by CRISPOR [14].

#### 3.1.3. Base and Prime Editors

As the potential for base editing as therapy for genodermatoses grows, a strong foundation for characterising its off-target effects is required. Although BEs are thought to share the same mechanism of Cas9-dependent off-target mutation formation, currently available prediction algorithms fail to efficiently recognise most likely edited sites [86,87]. Unlike basic CRISPR–Cas9, base editors possess an additional source of Cas9-independent off-target modifications—deaminases [88]. Genome-wide, seemingly random deaminations can be detected after treatment with cytosine base editors [89]. Alternative approaches to controlling both Cas9-dependent and independent off-target mutations came from the works of Rees et al. [90] and Gaudelli et al. [12].

Two studies in mice and rice demonstrated the viability of these alternatives and found that sgRNA-independent off-target editing has a predilection for highly transcribed regions of the genome and is more common in CBEs than ABEs [91,92]. Tools such as Endo-Digenome-seq21 and EndoV-seq [93] can successfully characterise sgRNA-dependent editing profiles for BEs, but they may underestimate sgRNA-independent editing. Whole-genome sequencing (WGS) methods may be effective for sgRNA-independent off-target assessment but are expensive, low-throughput and time-consuming [94]. Recently, methods employing rifampicin selection, orthogonal R loops, and kinetics assays have been validated for comparing sgRNA-independent off-target editing between CBE variants without using WGS [88]. These methods may be a basis for further work enabling the sensitive, cost-effective evaluation of sgRNA-independent off-targets in gene correction studies. Their unpredictable nature poses a great challenge, as the detection of single-nucleotide mutations caused by deaminases requires costly whole-genome sequencing and the careful interpretation of obtained data. Both CBE and ABE have also been shown to extensively deaminate nucleotides in RNA molecules, independently of DNA changes, and to cause considerable differences in gene expression and splicing [95,96,97]. Several modified deaminases have been developed to combat Cas9-independent off-target mutations: SECURE, RrA3F, AmAPOBEC1, PpAPOBEC1 and SsAPOBEC3B for cytosine base editors and SECURE-ABE for ABEs [96,98,99].

As a still-emerging technology, there are a paucity of studies on the off-target profile of prime editing; however, early data characterising its effects using ‘nDigenome-seq’ suggest that it is extremely precise [100]. A recent study demonstrated that PE3 produced undetectable pegRNA-independent off-target activity [101].

### 3.2. Designing Gene Editing Systems to Limit Off-Target Effects

Careful planning can mitigate off-target effects when deploying gene editing therapy. Web-based tools such as ‘CRISPick’ (Broad Institute) can be used to streamline the selection of sgRNAs based on the degree of their predicted off-target editing. However, selecting sgRNAs for CRISPR–Cas9 gene editing often involves maintaining an equilibrium between maximising on-target editing and off-target effects. For double-stranded cleavage to efficiently drive HDR repair and the excision of the mutation, the sgRNA must create a cut within approximately 30 nucleotides of the proximal ends of the dsODN donor template [102], which limits the choice of potential sgRNAs. Furthermore, the strict requirement of an ‘NGG’ trinucleotide PAM site downstream of the sgRNA limited the design of sgRNAs for base editing to only one or two possibilities, with no consideration of off-target editing potential. However, the emergence of base editors with Cas9 orthologs other than streptococcus pyogenes (sp), which recognise non-NGG PAM sites, enables greater flexibility for sgRNA choice.

In addition to sgRNA selection, one can use high-fidelity Cas9 (Cas9-HF1) nucleases, which are highly specific and cause almost undetectable levels of off-target editing [103]. A different approach that uses Cas9n to create single-strand staggered ‘nicks’ separately on both strands of the target DNA [104] led to reduced off-target editing following *COL7A1* gene correction in RDEB keratinocytes [40]. Delivery methods also impact off-target editing rates. The mRNA and RNP delivery of gene editors enables transient, controllable dosing in nuclei because these are quickly degraded, whereas plasmid or viral vectors can lead to higher off-target activity [88,105].

To mitigate base editor off-target editing, researchers have engineered new suites of CBEs, such as ‘YE1-BE424’ and ‘tCDA1EQ’, which offer 10–100-fold lower sgRNA-independent off-target editing than BE4 whilst largely maintaining on-target efficiencies [106]. In addition to their effects on the genome, an important study found that CBE and ABE cause the widespread deamination of RNA across the transcriptome [96]. These changes are independent of the sgRNA or any genomic mutations but could be diminished through pioneering ‘SECURE’ (Selective Curbing of Unwanted RNA Editing) CBE variants with specific mutations. Further work confirmed the transcriptomic editing profile of base editors and also developed ABE variants with lower proclivities to induce RNA edits for therapeutic applications [107]. To further reduce the risk of sgRNA-dependent off-target editing in base editing, it is additionally possible to use bubble hairpin sgRNAs, which reduce off-targets while maintaining a similar editing efficiency [108].

Base editing displays a further characteristic by which nucleotides adjacent and of the same type to the targeted mutation can be aberrantly converted in a process known as ‘bystander editing’ [109]. Bystander editing occurred in a couple of studies using ABE to correct *COL7A1* mutations in RDEB fibroblasts [14,70]. These undesired substitutions always arise within the base editor activity window and can therefore be easily detected when sequencing the region of interest using NGS. Furthermore, ‘BeHIVE’, a machine learning model, can be used to accurately predict base editing outcomes and determine the degree of bystander editing when designing experiments [110].

For both CRISPR–Cas9 and base editing, off-target editing remains a safety issue that can pose risks in clinical translation. Off-target mutations only need to be introduced in a small number of cells for potentially grave deleterious effects on patients to transpire; however, completely eradicating them is challenging. Important work has taken strides to address this by improving Cas9 and BE specificities and by enhancing off-target detection sensitivities. CRISPR–Cas9 has now been deemed sufficiently safe to enter over thirty clinical trials, and base editing has been featured in three trials by this year (although none yet for genodermatoses). Investigators designing CRISPR–Cas9 or base editing experiments for skin disease should follow the general principles discussed here to minimise off-target effects and deploy complementary in silico prediction algorithms and experimental tools in tandem to screen off-target editing.

## 4. Delivery

The delivery of gene editing tools to target cells is a fundamental requirement for gene editing therapies. Gene editing tools can be delivered as plasmids, mRNA, ribonucleic proteins (RNPs), or viral nucleic particles, each of which have varying advantages and disadvantages for specificity, efficiency and off-target effects. Delivering the editing tools can be achieved with viral vectors, injection, lipid nanoparticles, micro/nanoneedles or electroporation. Some of these tools can be used in vivo (Figure 3) or ex vivo on cells collected from a patient. Here, we describe first the available mechanisms of delivering gene therapy components into the cell, then describe how genetically modified cells can be used as ex vivo cell therapies for patients.

### 4.1. Electroporation

Electroporation is a method that efficiently delivers genetic payloads to cells by altering the cellular transmembrane potential (TMP). The TMP is created by different ion pumps and channels on the surface; when external electric pulses are applied, the TMP will increase, leading to a higher permeability of the cell as membranous pores form. The pores then allow cargo (e.g., nucleic acids) to pass through and enter the cell. Electroporation is now widely used in preclinical studies to deliver gene editing cargo into cells. For instance, Bonafont and colleagues used electroporation to deliver ex vivo dual sgRNA-guided Cas9 nuclease to delete *COL7A1* exon 80 in RDEB patient keratinocytes. Patient P2 and P1 cells analysed by NGS showed 95% and 87% exon 80 deletion, respectively [41].

Several advantages of electroporation have been demonstrated. Electroporation-based gene transfer can deliver a wider range of cargo to a variety of cell types [111] with a high efficiency. However, applied electric pulses will lead to local temperature increases that can denature cell surface proteins and limit normal cellular trafficking [112]. Electroporation can be cytotoxic and lead to necrosis and apoptosis [113,114]. These potential threats to the cell require a more in-depth analysis of its causation and more precise techniques to bypass these problems. In addition to ex vivo electroporation, gene editing tools can be directly injected into the dermis, which is then electroporated. This has been successfully conducted on RDEB mouse models in vivo as a proof of concept [115]. However, in vivo electroporation suffers from challenges relating to its invasiveness and small area of effect, which means that the method can currently be exclusively used for cells in an in vitro environment.

### 4.2. Viral Vectors

Viral vectors are currently the most popular choice for encapsulating gene editing tech during in vivo applications, with 35% of gene therapy clinical trials using them for delivery [116]. This is due to their low immunogenicity and long-term use for genetic transfections [117]. Although viral vectors cannot directly penetrate the intact stratum corneum, they can be applied to broken skin to reach target cells. This is the case for B-VEC, which uses herpes simplex virus 1 to deliver a full-length DNA copy of *COL7A1* into RDEB patients at wound sites [25]. B-VEC is currently in Phase III trials and is proving to be a promising next step in topical gene delivery.

Gene editing machinery can also be packaged into viral vectors and be injected into a target site for therapeutic delivery [25]. This has been achieved in vivo in humanized skin mouse models, in which CRISPR–Cas9 was delivered to remove exon 80 of *COL7A1,* resulting in the restoration of C7 expression and reduced skin fragility [22], thus demonstrating potential value for genodermatoses.

Despite their widespread use, viral vectors have several limitations. Viruses with a small immunogenic profile, namely adeno-associated viruses (AAVs), have a small payload size of ~4.7 Kb, meaning larger genes such as the 8.9 Kb *COL7A1* gene are unable to be fully encapsulated and thus making them unsuitable for gene addition. Cas9 and sgRNA are more appropriate for AAVs as they can be fully encapsulated, benefiting from a low immunogenicity [118]. Larger editing tools, such as base editor and prime editor, previously had to be dually administered when using AAVs, increasing the risk of immune responses that would target and degrade AAVs along with their therapeutic cargo [119]. Advances in ABE optimisation have since shown that it is possible to encode size-optimised ABE8e within a single AAV [119], improving the therapeutic potential of ABE.

Larger retroviruses such as lentiviruses (LVs) are used for larger cargo that AAVs are not suited for but cause a stronger immune response. This means that treatments utilising LVs as vectors are more likely to be degraded by immune cells [120]. This is especially relevant within the context of genodermatoses due to the population of Langerhans cells in the skin, which would increase the immune response [121,122]. Furthermore, multiple therapeutic administrations of LVs may trigger an adaptive immune response, reducing the efficacy of further treatment [123]. These drawbacks have caused investigators to seek novel non-viral vectors to overcome the challenges associated with viral vectors [124].

### 4.3. Non-Viral Nanoparticle Vectors

Non-viral vectors, particularly lipid nanoparticles (LNPs), are emerging as useful alternatives to viral vectors as they are less immunogenic and their modifiable nature allows them to fit most physical and payload size constraints [123]. These benefits make non-viral vectors attractive for in vivo and ex vivo genetic payload delivery (Figure 3).

The main challenge faced by non-viral vectors (particularly lipid-based vectors) is their susceptibility to degradation via the lysosomal pathway, therefore lowering drug efficacy as editing efficiency decreases due to rapid drug clearance [125,126]. This can be reduced in lipid-based vectors with the incorporation of lipids with added polyethylene glycol (PEG) into the nanoparticle (NP) structure. The PEG group is thought to create a ‘hydrophilic cloud’ around the NP, preventing cellular interaction proteins from recognising the NP, therefore preventing aggregation and integration into the lysosomal pathway and decreasing clearance [127,128]. It has been observed that some patients produce anti-PEG antibodies; however, these antibodies do not crosslink and are not associated with any pathology, and more investigation is required in this field [129].

Non-viral vectors are new to the field, so less research has been conducted in regard to their in vivo safety and efficacy. While non-viral approaches may one day replace viral vectors as the top gene editing delivery method, more in vivo research aimed at increasing editing efficiency is required before this can happen [124].

### 4.4. Micro/Nanoneedles

A highly novel delivery method that can be considered for the development of gene editing therapeutics for genodermatoses is that of micro/nanoneedle delivery. Silicon-based needles with a diameter of <50 nm can penetrate the stratum corneum through <50 nm diameter silicon-based needles [130], allowing for direct therapeutic delivery to the dermal layer [131].

Microneedles and nanoneedles trigger minimal cytotoxicity and have a low immunogenic profile due to the inert silicon used to synthesise nanoneedles. They also result in a high transfection efficiency, with a 90% transfection efficiency seen after 48 h using small interfering RNA (siRNA) to transfect HeLa cells in vitro [130], and they are appropriate for difficult-to-transfect cells [132]. The successful in vivo transfection of GFP-expressing DNA and labelled siRNA has been accomplished in mouse models [130]. Furthermore, as micro/nanoneedles must be directly applied to the region of desired editing, they are suitable candidates for in vivo therapy for genodermatoses [133,134]. This is particularly viable when considering the ease of synthesis in constructing nanoneedles compared with the considerations one must take when encapsulating gene editing components in viral/non-viral vectors; optimising nanoneedle manufacturing processes could mass-produce nanoneedles for general genetic transfection purposes [135]. Once completed, nanoneedles could be commercially bought and genetic payloads could be simply loaded into the needles for therapeutic/experimental use.

Microneedles have been successfully used to deliver CRISPR–Cas9 to mouse skin in inflammatory skin disease models such as atopic dermatitis to disrupt the *NLRP3* gene, thus increasing gene modification efficiency [136]. While extensive research if still required to demonstrate efficacy and safety in human patients, there is potential with microneedle delivery for painless in vivo therapeutics [115,135].

## 5. Ex Vivo Therapies

Many of the delivery tools discussed above can be used to deliver therapies in vivo (Figure 3)—however, when ex vivo approaches are considered, after the genetic editing of patient cells ex vivo, it is then necessary to introduce the cells back into the skin as a cell therapy. In genodermatoses, this has typically been conducted as injections or skin grafts.

### 5.1. Injections

Intradermal injections can be used to directly introduce gene-edited cells back into the skin [137]. The advantages of intradermal injections is that hospitalisation is not necessary and patients do not typically require anaesthesia [36]. This was performed in a Phase I clinical study on four adults with RDEB who received intra-dermal injections of modified autologous fibroblasts that were well-tolerated and led to a significant increase in C7, although no AFs were formed [138]. In another Phase I/II study by Marinkovich et al., five RDEB patients were recruited, and their biopsies were collected to extract fibroblasts. LVs were then employed to introduce a wild-type *COL7A1* gene to the fibroblast culture and sufficiently expand the fibroblasts for treatment. Finally, the genetically modified fibroblasts were injected into both blistering and intact skin. For 12 weeks, the injection sites were monitored. C7 expression was increased, and AFs were detected 3 months after injection, with 80% of wounds demonstrating good healing ability and no severe adverse events reported nor replicative viruses detected [139]. A similar principle could be followed after gene editing fibroblasts for therapy. However, the treatment was transient, as a 100% healing rate was observed at week 4 but declined to only 80% at week 12. This may have been due to the genetically modified fibroblasts entering senescence and failing to proliferate. For long-term treatment, this approach would need to be frequently applied.

Stem cells may be a valuable resource for injection gene editing therapy for genodermatoses. Mesenchymal stromal cells (MSCs) are multipotent, self-renewing stem cells that can be derived from a wide range of body tissues, such as bone marrow, adipose, and iPSCs [140,141]. MSCs also have immunomodulatory properties that regulate immune cells and can provide further benefits [142]. In a clinical trial, Rashidghamat et al. harvested allogenic bone marrow-derived MSCs (BM-MSCs) from healthy donors and administered these to 10 RDEB patients via intravenous infusion. Four of the nine patients developed encouraging increases in C7 expression, whereas five patients developed partial expression or no expression. The total blister count over the body of the participants decreased by 2.78 fold at day 28 and 2.88 fold at day 60 on average [143]. However, the selection and extraction of donor MSCs is a complex and robust process that requires extensive regulatory oversight. Before transplantation, cells need to be screened against any infectious diseases and the genomic DNA needs to be checked to see if the gene associated with the disease is mutation-free [143]. MSCs can also be delivered through topical applications, which are a less invasive method that has been shown to improve wound healing and skin grafting survival rate following surgery [144]. These challenges could be ameliorated with the use of autologous, genetically modified MSCs, which may provide a promising future avenue.

### 5.2. Grafting

The skin grafting of patient-derived, gene-corrected cells is another avenue for ex vivo therapies. This could be achieved with holoclones, somatic stem cells that generate keratinocytes in the epidermis [115]. DrozGeorget Lathion et al. transfected epidermal stem cell holoclones with a replicative defective retrovirus containing full-length *COL7A1* complementary DNA. Cells were then transplanted into immunodeficient mice to form a non-blistering epidermis. The results showed that many anchoring fibrils self-renewed, and C7 was found in the basement membrane using an electron microscope [116]. A standout example of skin grafting was performed by Hirsch et al., who treated a 7-year-old patient with JEB that was caused by mutations in *LAMB3* [145]. They collected biopsies from the patient’s inguinal region and used retroviral vector-based therapy to add a full-length *LAMB3* cDNA to the patient-derived keratinocytes. The keratinocytes were grown into skin grafts that were applied to the patient. At 21 months post-engraftment, 80% of the patient’s total body surface area was restored. Even under mechanical stress, no blisters or erosion was observed [145]. This regeneration of almost the entire epidermis was sustained by transgenic holoclone-derived cells [146], demonstrating that the skin grafting of these cells is a promising approach for ex vivo genetically engineered cell delivery.

Skin grafting presents several advantages, as epidermal skin grafting does not usually leave scars during application and patients can therefore heal quickly [147]. This prevents the creation of secondary injuries to the patient. General anaesthesia is also not required for skin grafting, which makes it more cost- and time-effective [148]. However, limitations do exist. Graft failure is a threat to patients, especially in the case of large-area grafting, which may cause severe infection and issues with blood circulation. Additionally, skin grafting is usually a last step solution for severe skin disorders, as previously described [145], and the process to grow transgenic skin grafts is extensive and time-consuming.

## 6. Animal Models

Finally, a brief note about animal models. Mouse and similar mammalian models are most commonly used for genodermatoses research due to their similarity in skin structure, genome, and relevance for drug testing. Many mouse models have been developed, especially for dystrophic EB (DEB). As new treatment strategies for gene therapies, cell therapies, and protein therapies are developed, it is crucial to demonstrate efficacy in animal models prior to clinical testing. Transgenic mice are particularly valuable for demonstrating the efficacy of gene editing therapies.

The first DEB model mouse was generated by Heinonen et al. in 1999 by knocking out *Col7a1* [149]. This resulted in severe blistering below the lamina densa and a complete absence of anchoring fibrils. However, the mice died during the first two weeks of life due to the severity of the phenotype, posing a challenge for testing therapeutics. In 2008, a new mouse model with hypomorphic *Col7a1* expression was described [150]. This mouse grew to adulthood and modelled severe RDEB with only 10% wild-type C7, which would allow for the more in-depth pathophysiological modelling of RDEB [151]. Furthermore, to investigate the role of the immune system of the RDEB phenotype, NOD/SCID IL2rγc^null^ (NSG) mice embryos were edited with CRISPR–Cas9 to knock out *Col7a1*, resulting in a milder phenotype than seen in previous mouse models. Furthermore, the immunodeficiency allowed for the trial of human cell therapies in mice models without triggering an immune response, which would prove valuable for ex vivo gene therapies [152]. For gene editing therapies, it is also important that the mutations in mouse models match mutations found in human patients, which was achieved with DDEB mutations in 2021 [153]. In the future, this would allow gene editing technologies to be trialled on mouse models as part of pre-clinical work to see the systemic effects of gene editing tools. Currently, gene editing therapies are trialled on immunodeficient mouse models with human skin equivalents xenografted onto them, allowing for studies to be conducted on human cells supported by mouse skin tissues and other organs.

While non-human primates would be well-suited for modelling the systemic effects that accumulate in genodermatoses, none are currently available. Macaques with homozygous *KRT5* mutations have been identified; however, those were stillborn, likely due to the severe EBS morphology. The studied heterozygous macaques did not have EBS [154]. As such, these primates are not appropriate for gene editing studies.

Other animals, such as zebrafish and Drosophila, have also been used to model genodermatoses. This may be achieved with knockdowns, such as those of *col17a1* homologues, which are relevant in JEB pathology [155]. *Snap29* mutant zebrafish larvae have shown skin defects comparable to cerebral dysgenesis, neuropathy, ichthyosis and keratoderma (CEDNIK) disease caused by the homologous *SNAP29* gene variants in humans [156]. Beyond vertebrates, Drosophila has also been used to model EBS [157], demonstrating similar keratinization networks and blistering to patients. However, these animal models are less suited for trialling gene editing therapies than mouse models due to the genomic differences between the species.

In the future, mouse models would ideally be developed with mutations in a wider range of genes that cause genodermatoses, such as *Lamb3* or *Krt14*. A repository of mice with precise disease-causing mutations, as achieved in DEB, would allow for the most precise testing of specific gene editing technologies possible in pre-clinical trial testing. For now, xenografted mice models with human skin equivalents remain the gold standard for gene editing testing.

## 7. Conclusions and Future Directions

Gene editing therapies have yet to reach the clinical setting for genodermatoses, but recent work in both patient-derived cells and animal models has shown promise, thus far focussing on the rare and severe forms of EB. Challenges regarding gene correction efficiency, off-target safety, and delivery methods remain; however, the continuous advancement of better, more precise gene editors in all areas of medicine will only benefit the field. It will also be essential to optimise different gene editing tools for different cell types due to the difference in mitotic and post-mitotic cells. Mouse studies looking at the long-term outcomes of gene editing will become more necessary for pre-clinical studies. Fortunately, gene replacement therapies are providing a blueprint, and using the lessons learned from them will allow for the rapid development of clinical trials for gene editing in genodermatoses, especially approaches for delivery and the investigation of off-target safety concerns. Gene editing therapies for other diseases have begun to enter the clinical setting, which will provide further insight into gene editing for use in genodermatoses. The first clinical trials involving gene editors for genodermatoses will undoubtably occur soon and offer hope for a permanent cure for people living with genetic skin disorders.

## Figures and Tables

**Figure 1 ijms-24-02298-f001:**
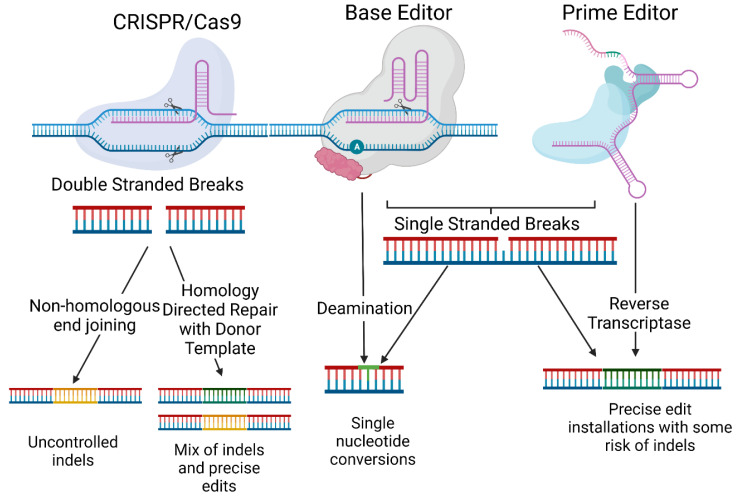
Comparison of CRISPR–Cas9, base editor and prime editor. CRISPR–Cas9 causes double-stranded breaks, resulting in uncontrolled indels when paired with non-homologous end joining. When used with a donor template, this can trigger the homology-directed repair pathway, resulting in a mix of indels and precise edits. Base editor and prime editor both generate single-stranded breaks. Base editor combines this with deamination and mismatch repair to generate single nucleotide conversions. Prime editor instead uses a reverse transcriptase to encode many possible types of precise edits. Figure was created in BioRender.

**Figure 2 ijms-24-02298-f002:**
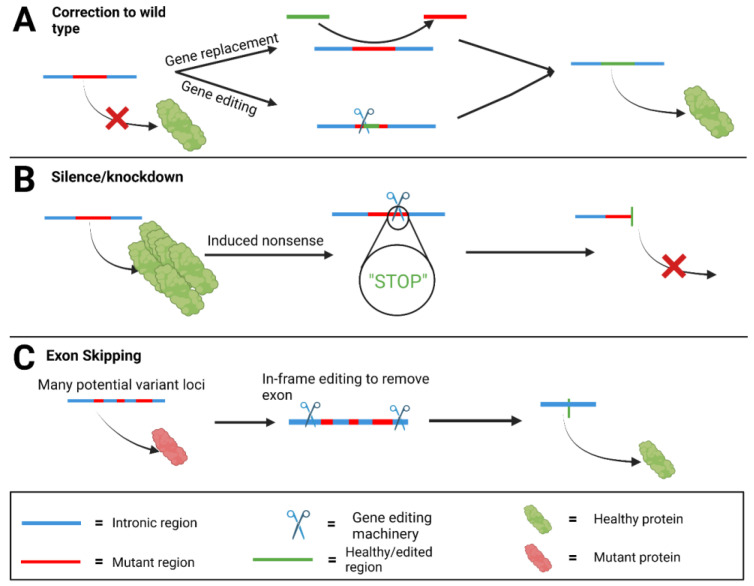
Overview of correction approaches to genetic disorders. Mutated genes and proteins (red) can be treated through several genetic correction methods, creating less harmful genes and proteins (green). (**A**): Mutant genes can be corrected to the wild type using gene editing tools or gene addition therapy. (**B**): Hypermorphic mutations can be silenced through the induction of a ‘STOP’ codon. (**C**): Where there are in-frame exons of a gene of which many possible disease-causing variants have been identified, the entire exon can be removed with gene editing to restore healthy protein function. Created with BioRender.

**Figure 3 ijms-24-02298-f003:**
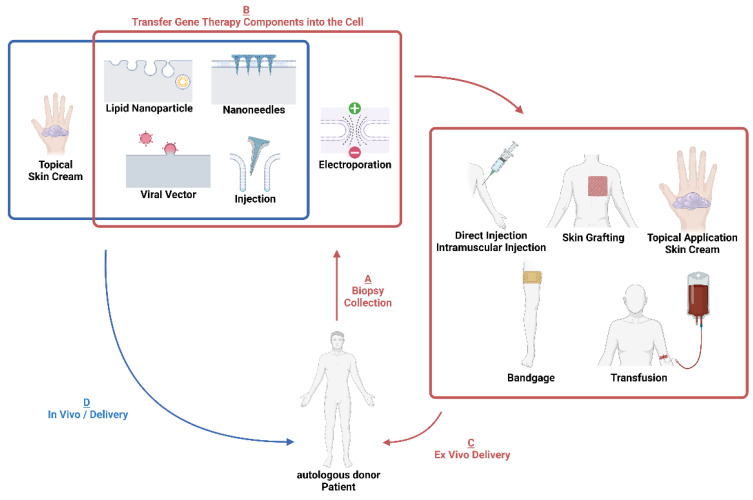
Graphic overview of ex vivo and in vivo delivery systems for therapeutic gene editing. Ex vivo delivery is highlighted in red on the right. A: biopsy is collected from patients and target cells are purified for cell culturing in vitro. B: gene editing tools are transferred into the cells to genetically modify the genome to restore normal function. C: the edited cells are then transferred back to the patients through different routes. In vivo delivery is highlighted in blue on the left. D: gene editing cargo is packed in delivery intermediates and then transferred to patients’ target tissue to finish editing. Created with BioRender.

**Table 1 ijms-24-02298-t001:** Gene therapy trials in genodermatoses as listed on ClinicalTrials.gov (accessed on 29 November 2022) with reported results or in progress.

Disease	Therapeutic	Gene Therapy Delivery Method	Phase	Outcome	Reference
Netherton syndrome	Autologous skin sheets containing additional SPINK5 gene	Lentiviral Vector	I	Transient functional correction in 1 Patient	[23]
Recessive dystrophic epidermolysis bullosa (RDEB)	Autologous skin sheets containing additional *COL7A1* gene	Retroviral Self Inactivating	I/II	In progress	[33]
RDEB	Autologous epidermal sheets containing additional *COL7A1* gene	Retroviral	I/II	Favourable safety and efficacy outcomes—Phase III in progress	[34]
DEB	Topical beremagene geperpavec (carries HSV1-COL7) applied to wounds	Self-inactivating HSV1	I/II	Durable wound closure with minimal adverse events—Phase III in progress	[25]
Autosomal recessive congenital ichthyosis	Topically administered KB105 containing *TGM-1*	Self-inactivating HSV-1	I/II	In progress	[35]
junctional epidermolysis bullosa	Epidermal autograft containing *LAMB5*	Gamma-retroviral	II/III	In progress	[24]
RDEB	Intradermal Injections of *COL7A1*-modified autologous fibroblasts	Self-inactivating lentivirus	I	Increased C7 observed after 12 months but no mature anchoring Fibrils	[36]

## Data Availability

Data sharing not applicable.

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
