# Peer review of "Challenges of Gene Editing Therapies for Genodermatoses"

_ijms, 2023, doi:10.3390/ijms24032298_

Round 1

Reviewer 1 Report (Previous Reviewer 2)

In the manuscript titled “Challenges of Gene Editing Therapies for Genodermatoses” Brooks et al make a very complete review of advances in gene therapies for skin diseases. The manuscript is well written and only few errors were observed.

I have a few comments as following:

1.     Include information on the different software for the design of the most efficient guides.

2.     Include information on PAM-Les CAS9 ans PAM-Less Base editor (in vitro and in vivo). Walton, R. T., Christie, K. A., Whittaker, M. N., & Kleinstiver, B. P. (2020). Unconstrained genome targeting with near-PAMless engineered CRISPR-Cas9 variants. Science368(6488), 290-296.

3.     In the Animal Models section, include non-human primates. If there is any work.

Author Response

Response to Reviewer 1

In the manuscript titled “Challenges of Gene Editing Therapies for Genodermatoses” Brooks et al make a very complete review of advances in gene therapies for skin diseases. The manuscript is well written and only few errors were observed.

I have a few comments as following:

Answer: Thank you for your positive and encouraging feedback! We appreciate your comments which we have addressed further.

  1. Include information on the different software for the design of the most efficient guides.

      Answer: Thank you for this improvement upon our review. We have detailed some of the different software for CRISPR-Cas9, Base editing and Prime editing in section 2.2, line 165. This includes CHOPCHOP, CRISPOR, BE-Designer, BEditor, Prime Design and BnP Designer (reference 44-49).

  1. Include information on PAM-Les CAS9 ans PAM-Less Base editor (in vitro and in vivo). Walton, R. T., Christie, K. A., Whittaker, M. N., & Kleinstiver, B. P. (2020). Unconstrained genome targeting with near-PAMless engineered CRISPR-Cas9 variants. Science368(6488), 290-296.

Answer: Thank you for this suggestion. We have added a small paragraph on PAM-less tools in the introduction, line 78 (reference 17).

  1. In the Animal Models section, include non-human primates. If there is any work.

Answer: Thank you for this suggestion! We have included the only primate model published, Macaques with EBS, however, these were stillborn and thus don’t currently have any use for gene editing therapies (reference 154). This is described on line 634.

Reviewer 2 Report (Previous Reviewer 3)

The review has substantially improved in quality and focus. 

Here are some minor mistakes to correct  at the corresponding pdf text lines

Line 92: delete the word shift after in-frame

Line 431: include COL7A1 before exon 80

Line 455: Include a mention and the corresponding reference (new reference) to the recent study by García et al. about in vivo gene editing of COL7A1 with an adenoviral vector (Mol Ther Methods Clin Dev. 2022 Sep 16;27:96-108.).   

Line 500: change the word needs for needles

Line 578: holoclone-derived cells is more appropriate than holoclone cells

Additional comment: reference to Clinical trial phases is sometimes in roman numbers and others in arabic numbers. Homogenize.

Author Response

The review has substantially improved in quality and focus. 

Here are some minor mistakes to correct  at the corresponding pdf text lines

Line 92: delete the word shift after in-frame

Line 431: include COL7A1 before exon 80

Line 455: Include a mention and the corresponding reference (new reference) to the recent study by García et al. about in vivo gene editing of COL7A1 with an adenoviral vector (Mol Ther Methods Clin Dev. 2022 Sep 16;27:96-108.).   

Line 500: change the word needs for needles

Line 578: holoclone-derived cells is more appropriate than holoclone cells

Additional comment: reference to Clinical trial phases is sometimes in roman numbers and others in arabic numbers. Homogenize.

Answer: Thank you very much for the positive feedback. We are very grateful to be alerted of the following corrections. We have made the minor corrections recommended, as can be seen on lines 98, 447, 472, 518, 596. The García reference has been included (reference 22) to expand on our point about AAV and to replace the LCA example.  Additionally, we have homogenized the clinical trial phases to roman numerals. Thank you for bringing these mistakes to our attention so we can improve our manuscript.

This manuscript is a resubmission of an earlier submission. The following is a list of the peer review reports and author responses from that submission.

Round 1

Reviewer 1 Report

Excellent and comprehensive review addressing all points of gene editing such as efficiency, methods and delivery. The senior author has an excellent academic record and high scientific standards.

One point would be to ensure all acronyms are spelled out when first mentioned in the text as there are quite a few. 
the figures are simple and easy to understand. It is a highly educational review summarising current state of play for gene editing in genodermatoses. 

Reviewer 2 Report

In the manuscript titled “Challenges of Genes Editing Therapues for Genodermatoses an beyond”, Brooks et al. focus their work on the study of skin diseases and possible treatments and therapies. The manuscript is well written and only few errors were observed. The following concerns should be addressed before the manuscript is considered to be published.

1. Authors could describe the animal models (mouse and zebrafish) that exist. What type of mutations do these models have, whether or not they are similar to those of the patients described? And what efficiency of correction do they show after therapy.

2. The authors could detail other editing tools such as TALENs

3.  It would be very interesting to see a table or figure of the treatments that are being used and the phase of clinical development in which they are.

Reviewer 3 Report

The review of Brooks et al., is a text that seems to be produced by someone with very little or no experience in science writing and little or no critical reading by a senior advisor. Besides, mistakes not only concern to the formal aspects but also to the specifics of the review subject.

Following, I pinpoint only a limited number of the inaccurate or incorrect statements along the manuscript.

Since the beginning of the introduction, the text contains defects: Authors refer to genodermatosis by listing four random entities such as EB, Harlequin Ichthyosis, KLICK syndrome or Xeroderma Pigmentosum without even making a brief mention of the classification of these diseases. Why mixing general forms (e.g. EB) with specific forms (e.g. Harlequin Ichthyosis)? Why Harlequin Ichthyosis and not a more prevalent form (e.g. Lamelar ichthyosis)?

A little later in the introduction they refer to clinical trial stages instead of phases.

They then inaccurately comment that the insertion (of the transgene) causes mutations in other genes when, as it has been seen in most cases, insertion results in the overexpression of a potential oncogene by activation of its promoter sequences. They then refer to the non-physiological expression levels of the transgene discussing about silencing and not mentioning overexpression that is more relevant.

Continuing in the introduction, when they refer to the use of CRISPR/Cas9 and donor template for homologous recombination, they omit recent work of others in which the recombination efficiency is high.

They then contradict themselves saying that Prime editing has been tested on RDEB but has not been tested on the context of genodermatoses.

In the section “Approaches in gene editing” the list of inaccuracies/omissions continues: The subtitle is already confusing; “Correction to wild type”. Something like “Restoration of the wild type gene sequence” may be more appropriate.

In this section authors begin by naming HDR and omit the NHEJ.

This section closes with a reference without the slightest thread to the correction of a gene that causes a myopathy.

In the following section “Knock out /silencing, they refer to hypermorphic when the most appropriate manner is to refer to a dominant trait. Following, they jump without the slightest sense to the correction of hypercholesterolemia. Then, they mention repeatedly that this strategy would be used when restoring the wild type sequence is impossible.

In the section on “activation of a less harmful gene”, authors thoroughly describe the correction of the globin genes but comment that no genodermatosis susceptible to this approach is known. Why getting into this business then?

In the next section “Gene editing efficacy” the untidiness continues.

In a sentence that does not end, authors say that high efficiency is achieved in some methods when the cells are selected for. Selected for what?

They go on to exhaustively describe ways to improve HDR but make no mention of any application to genodermatoses. Below, they describe in detail the base and prime editing technologies, as if the review had a major focus on those and not on their application.

In the following section “improving efficiency…” there is an important error: the concept of working under GMP conditions being associated to the impossibility of using markers such as GFP or cell sortin. This denotes ignorance about the subject.

The following section is again a treatise on off targets but with minimal, if not null, reference to studies in genodermatosis.

One of the most serious mixed-up concepts is the one corresponding to point 5 (i.e delivery) where what one expects is that the literature on the delivery of editing systems be reviewed and we find that what is being discussed is the delivery of genetically modified cells not even specifically through genome editing (5.1 injections and 5.2 grafting). In the end, authors refer with a little more correction to the use of electroporation, viral and non-viral vectors, although the latter not very focused on gene editing in genodermatosis.

In a quick check of the bibliography, a repetition of reference 14 (Izmiryan et al.) was detected. That obviously affects the order of references.

All in all this is a very muddled piece of work that should not be accepted for publication.